# Broiler Spaghetti Meat Abnormalities: Muscle Characteristics and Metabolomic Profiles

**DOI:** 10.3390/ani14081236

**Published:** 2024-04-20

**Authors:** Teng Wu, Pingping Liu, Jia Wu, Youluan Jiang, Ning Zhou, Yang Zhang, Qi Xu, Yu Zhang

**Affiliations:** 1College of Animal Science and Technology, Yangzhou University, Yangzhou 225009, China; tengwu2022@163.com (T.W.); 13525983362@163.com (P.L.); wj18252789252@163.com (J.W.); j25802580128@163.com (Y.J.); znznzn20001231@163.com (N.Z.); zyang@yzu.edu.cn (Y.Z.); xuqi@yzu.edu.cn (Q.X.); 2Key Laboratory for Evaluation and Utilization of Livestock and Poultry Resources (Poultry), Ministry of Agriculture and Rural Affairs, Beijing 100176, China

**Keywords:** broiler, spaghetti meat, muscle characteristic, metabolomics

## Abstract

**Simple Summary:**

Broilers are important poultry, and recent muscle abnormalities in broilers, such as the emergence of spaghetti meat (SM), have posed challenges to the poultry industry. This condition has received less research attention compared to similar abnormalities, such as wooden breast and white striping. Therefore, in this study, we systematically compared the differences in appearance, meat quality, muscle fiber characteristics, and metabolic components between SM and normal tissues. The results showed that compared with the normal group, the SM group had softer meat quality, and significantly higher yellowing and brightness, while pH and protein levels were significantly lower. Key differential metabolites, including 14,15-DiHETrE and L-malic acid, as well as key metabolic pathways related to inflammation and oxidative stress, such as linoleic acid, arachidonic acid, phenylalanine, and histidine metabolism, are associated with SM. This study provides valuable insights into the muscle characteristics of SM and helps identify potential mechanisms and nutritional regulatory targets of SM myopathy.

**Abstract:**

Spaghetti meat (SM) is a newly identified muscle abnormality that significantly affects modern broiler chickens, consequently exerting a substantial economic impact on the poultry industry worldwide. However, investigations into the meat quality and the underlying causative factors of SM in broilers remain limited. Therefore, this study was undertaken to systematically evaluate meat quality and muscle fiber characteristics of SM-affected meat. To elucidate the disparities between SM-affected and normal (NO) muscles in broiler chickens reared under identical conditions, we selected 18 SM-affected breast tissues and 18 NO breast tissues from 200 broiler chickens raised according to commercial standards under the same conditions for our study. The results showed that compared with the NO group, the muscle surface of the SM group lost integrity, similar to strip and paste. The brightness and yellowness values were significantly higher than those of the NO group. On the contrary, the shear force and protein were significantly lower in the SM group. Microscopic examination revealed that the muscle fibers in the SM group were lysed, necrotic, and separated from each other, with a large number of neutrophils diffusely distributed on the sarcolemma and endometrium. Thirty-five significantly different metabolites were observed in the breast muscles between both groups. Among them, the top differential metabolites—14,15-DiHETrE, isotretinoin, L-malic acid, and acetylcysteine—were mainly enriched in lipid metabolism and inflammatory pathways, including linoleic acid, arachidonic acid, phenylalanine, and histidine metabolism. Overall, these findings not only offer new insights into the meat quality and fiber traits of SM but also contribute to the understanding of potential mechanisms and nutritional regulators for SM myopathy.

## 1. Introduction

In recent years, there has been a noticeable increase in the occurrence of breast muscle abnormalities, including white striping (WS) and wooden breast (WB) in broiler chickens [1]. A newly identified muscle abnormality, known as spaghetti meat (SM), has also emerged, which has adversely affected the poultry industry [2]. Studies have indicated that the prevalence of SM can be as high as 20%, leading to adverse consequences on the composition, nutritional quality, and apparent attributes of meat [3]. This, in turn, results in a considerable reduction in meat value and significant economic losses owing to the need to downgrade affected meat products [4]. Despite its economic impact, the specific characteristics of this muscular abnormality and the underlying mechanisms remain partially understood and have received relatively less research attention than other similar conditions, such as WS and WB.

Heterogeneous chicken breasts have low acceptability and are generally used to produce highly processed products [5]. However, owing to suboptimal protein functional properties, water retention capacity, and textural properties, the quality of these products is compromised. This simultaneously leads to a reduction in their nutritional value to a certain extent. For example, WS is characterized by higher lipid levels and significantly lower ash content [6], whereas WB-affected breast meat exhibits increased weight and pH, fat, and connective tissue content, as well as lower protein levels [7,8]. Meat affected by SM exhibits a visual resemblance to spaghetti and is characterized by quality deficiencies, such as high pH, poor water retention, and low nutritional value.

However, there is currently no consensus on the definitive quality characteristics of this form of heterogeneous meat, and the underlying mechanism of SM remains elusive [9]. Oxidative stress and metabolic abnormalities are considered important causes of abnormal muscle fiber development. Oxidative stress can damage protein function, chemically modify certain lipids and nucleic acids, and stimulate the decomposition of muscle fibers, resulting in muscle atrophy [10]. Perturbations in the levels of specific differential metabolites may induce dysfunctions and textural lesions [11]. Moreover, varying responses of hormones that regulate metabolism, protein synthesis, intramuscular fat deposition patterns, and intracellular signaling, including insulin growth factor, growth hormone, and myostatin, may play a role in the formation of heterogeneous meat. Compared to normal (NO) meat, SM-affected meat exhibits a significant 10% reduction in protein content, alongside a notable increase of 21.8% and 3% in fat and moisture contents, respectively. Moreover, SM-affected broiler chickens exhibited significantly reduced eicosapentaenoic acid, docosahexaenoic acid, and hydroxylysylpyridinoline contents, and a large number of glycosaminoglycans was observed in the perimuscular connective tissue [12]. Notably, the incidence of SM is higher in female chickens. Analyzing the tissue morphology and quality of SM-affected meat, along with employing metabolomics to determine the components and key regulatory pathways, holds the potential to facilitate the establishment of assessment criteria and reveal potential underlying mechanisms. Furthermore, this approach may serve as a basis for alleviating the problems associated with SM through future endeavors in genetic breeding and nutritional regulation, ultimately providing substantial evidence for its occurrence. 

In this study, we conducted a systematic evaluation of meat quality characteristics (pH value, shear force, protein and fat contents, and other indicators) and muscle fiber characteristics of SM-affected meat. Additionally, we employed targeted metabolomics to analyze the differential metabolism in SM-affected meat. This involved tracing specific substances and analyzing key metabolic pathways to establish the assessment criteria for SM and explore the biomarkers and regulatory pathways associated with its occurrence.

## 2. Materials and Methods

### 2.1. Ethics Statement

All animal experiments were approved by the Institutional Animal Care and Use Committee of Yangzhou University (Approval Number: 151-2018). The procedures were strictly implemented following the “Regulations on the Management of Laboratory Animal Affairs” (Yangzhou University, Yangzhou, China, 2012) and the “Experimental Practice Management Standards” (Yangzhou, China, 2008).

### 2.2. Animals and Feeding Management

This study included 200 42-day-old Ross 308 broiler chickens (all males). They were subjected to a scientifically formulated feeding regimen, receiving chicken feed from 0 to 42 days of age with consistent nutritional levels and feeding environment. The nutritional level of the basic diet met the established nutritional requirements of local Chinese chickens (NY/T33-2004 [13]). The premix used in the diet met the requirements of the National Research Council for broiler chickens. Detailed information on the composition and nutritional content of the basal chicken diets is provided in Appendix A.

### 2.3. Sample Collection

The experimental samples were collected from Jinghai Poultry Industry, Nantong, Jiangsu, China. The feed was withdrawn 10 h before slaughter but the water is available for chickens. The 42-day-old Ross 308 broiler was slaughtered according to standard industrial practices. We used voltage (220 V, 50 Hz) to shock the broiler chicken into a coma, cut the throat, and soaked it in 65 °C water, then used a machine to depilate it, remove internal organs, and refrigerate them on ice. Subsequently, two trained investigative team members categorized the breast muscles into NO and SM groups (each group, *n* = 18) by visual inspection and chest palpation methods [14]. Among them, 18 deboned unilateral breast muscles showed spaghetti shape with poor tear toughness and belonged to the SM group. Meanwhile, 18 of the 182 breast muscles without visible abnormalities to the naked eye were randomly selected to belong to the NO group. The muscles were trimmed of obvious adipose tissue and connective tissue and then stored on ice at 4 °C until the measurements of meat quality parameters. We collected three samples from each NO and SM breast muscle. One part was stored at 4 °C for meat quality determination, one part was used for microscopic sectioning in preservation solution, and the other part was stored at 70 °C for nontargeted metabolomics detection.

### 2.4. Analysis of Physical Properties and Proximate Composition

Broiler breast meat samples were subjected to a comprehensive analysis, including assessment of pH, color, cooking weight loss (CWL), shear stress, moisture levels, and protein content. The breast muscles were specifically separated to analyze for meat color, CWL, and texture. Color parameters, namely, lightness (L*), redness (a*), and yellowness (b*), were determined according to the color system established by Zhang et al. using a colorimeter (CR-400, Konica Minolta, Tokyo, Japan) [15]. CWL was evaluated by weighing muscle samples wrapped in aluminum paper and cooked on a baking sheet at 150 ± 5 °C using a semianalytical scale. After reaching 35 °C, the samples were turned over and left to cook until an internal temperature of 72 ± 2 °C was reached. The percentage difference between the initial and final sample weights was calculated, representing the CWL. The water, protein, fat, and collagen contents of the muscles were measured using a meat component rapid analyzer (Food Scan type, Shanghai Mettler-Toledo Instrument Co., Ltd., Shanghai, China) [16].

### 2.5. Analysis of Muscle Fiber Morphology

Following the slaughter of broiler chickens, the NO and SM breast muscles selected through visual observation and touch, and the fascia and epidermis at the top of the left pectoralis major muscle were removed. Samples of the pectoral muscle tissue, aligned parallel to the muscle fibers (2 cm × 1 cm × 0.2 cm), were fixed in 4% paraformaldehyde fixative (Solarbio) for 48 h. Frozen samples were cross-sectioned and stained with hematoxylin and eosin (Khaliq, Beijing, China, 2023). Subsequently, samples were scanned using a Nano Zoomer scanner (Hamamatsu, Sydney, Australia). Images were acquired under the same conditions and magnification. Four fields of view were obtained per section, and three sections were acquired per sample. We used a microscope to observe the morphology of NO and SM muscle fibers.

### 2.6. Liquid Chromatography with Tandem Mass Spectrometry Untargeted Metabolomic Analysis

For liquid chromatography with tandem mass spectrometry analysis, sample pretreatment steps were performed according to a method described by Navarro-Reig et al. [17]. A Thermo Vanquish ultra-HPLC system (Thermo Fisher Scientific, Waltham, MA, USA) equipped with a Vanquish liquid chromatograph (2.1 × 150 mm, 1.8 µm) and a Thermo Orbitrap Exploris 120 mass spectrometer (Thermo Fisher Scientific, USA) was used for liquid chromatography with tandem mass spectrometry analysis. The mobile phase included 0.1% formic acid acetonitrile (C) and 0.1% formic acid water (D) in positive ion mode and 5 mmol/L acetonitrile (A) and 5 mM ammonium formate water (B) in negative ion mode, with a flow rate of 0.25 mL/ min. The elution gradient settings were as follows: 2% C, 1.5 min; 2% to −50% B, 9 min; 50% to 98% C, 12 min; 98% C, 13 min; 98% to 2%, 14 min, 2% C, 15.5 min [18]. The precision Thermo Orbitrap Exploris 120 mass spectrometer was operated in positive/negative polarity mode, with a spray voltage of 3.50 kV, sheath gas flow rate of 30 arb, auxiliary gas flow rate of 10 arb, and capillary temperature of 325 °C [19].

### 2.7. Statistical Analysis

Data analysis was conducted using Excel 2019 software. SPSS software (version 22.0) was used to compare the average values of various meat quality detection indicators in the NO and SM groups using the independent sample *t*-test. *p* < 0.05 was considered statistically significant. Results are expressed as “mean + SD”. The R software package Ropls [20] was used to generate score plots, loading plots, and S-plots based on the sample data, illustrating the differences in metabolite composition between samples. The original mass spectrum offline files were converted into mzXML file format using the MSConvert tool in the ProteoWizard software package (v3.0.8789) [21]. Metabolite identification was performed by matching accuracy mass (<30 ppm) and tandem mass spectrometry data with entries in HMDB [22] (http://www.hmdb.ca, accessed on 10 January 2024) and KEGG (Ogata et al., 1999) (http://www.genome.jp/kegg/, accessed on 10 January 2024) databases. The differential metabolites were subjected to pathway analysis using MetaboAnalyst [23], which combines results from robust pathway enrichment analysis with pathway topology analysis.

## 3. Results

### 3.1. Comparison of Physical Properties and Proximate Compositions

The phenotypic assessment of the breast muscles between the NO and SM groups showed that the breast muscles in the SM group exhibited a glossy, yellow surface, with some areas appearing white (Figure 1). Tactile examination revealed that the SM-affected muscle bundles were loose, and the breast muscles exhibited a soft texture. Applying pressure to the muscle surface revealed a soft and sticky quality, and the meat was susceptible to tearing by hands. The strip-shaped muscles are clearly exposed on the surface of SM. Muscles are easily torn apart and fiber toughness is relatively fragile. There were significant differences in brightness, yellowing, pH, shear force, and protein expression between the NO and SM groups (Table 1). The brightness and yellowing values of the SM group were significantly higher than the NO group, while the pH value, shear force, and protein content of the NO group were significantly higher than the SM group. However, there was no difference in unilateral breast weight and carcass weight between SM and NO.

### 3.2. Comparison of Muscle Fiber Morphologies

Compared to the NO group, the SM group exhibited severely affected muscle development, showing a trend of muscle degeneration with notable alterations in muscle morphology and phenotype (Figure 2). Histological examination of the hematoxylin and eosin-stained sections of the breast muscles showed that the NO group exhibited typical polygonal muscle fibers in the cross-section, with clearly delineated endomysial gaps between the muscle fibers, well-defined perimysial lines, and no observable inflammatory cells (Figure 2A). Conversely, in the SM group, the muscle fibers appeared denser, with narrower gaps between the muscle cells, and most muscle fibers took on a flat, elongated strip-like appearance. A large number of neutrophils were observed on the perimysium, which were densely distributed on each perimysium, forming clusters and, in some instances, infiltrating the endomysium (Figure 2B). Furthermore, the diameter of SM-affected fibers was significantly larger than that of NO fibers (*p* < 0.01) (Figure 2E). Compared to the NO group, the perimysium in the SM group was scattered and exhibited a larger distribution area. The SM sections showed loose muscle fibers and larger gaps between them (Figure 2C). Inflammatory cells were scattered around the perimysium, and membrane fibrosis was also observed (Figure 2D).

### 3.3. Identification of Differential Metabolites in SM 

#### 3.3.1. Screening of Differential Metabolites

The screening of significantly different metabolites between the NO and SM groups was conducted, employing variable importance in projection score > 1 and *t*-test *p* < 0.05 as criteria [24]. Upregulated metabolites are indicated in red, whereas downregulated metabolites are indicated in blue (Figure 3A). The statistical representation of the differential metabolites between the SM and NO groups is shown in Table 2. There were 35 significantly different metabolites in the breast muscles. There were 17 upward and 18 downward adjustments between the SM and NO groups.

#### 3.3.2. Multivariate Analysis

Multivariate analysis was performed to assess the metabolic spectrum differences between the NO and SM groups. The corrected partial least squares discriminant analysis score plot showed a clear distinction between the NO and SM groups (Figure 3B). Orthogonal corrected partial least squares discriminant analysis of the chromatographic data further highlighted distinctions in the meat quality metabolites between the NO and SM groups (Figure 3C). 

#### 3.3.3. Hierarchical Clustering Analysis of Differential Metabolites

Differential metabolites, such as 14,15-DiHETrE, isotretinoin, capsidiol, acetylcysteine, and 1-arachidonoylglycerol, were significantly upregulated (Figure 3D). The hierarchical clustering diagram revealed 35 differential metabolites (Appendix A). The relative content in the figure is indicated with different colors. The redder the color, the higher the expression level, and the bluer the color, the lower the expression level. The columns and rows represent samples and metabolite names (Figure 3E). Among them, most in the SM group were organic compounds. Notably, 17 metabolites were more abundant, primarily comprising lipids, lipid molecules, and organic acids and their derivatives. Seven of these metabolites were lipids and lipid-like molecules, including dodecanoic acid, capsidiol, 2-trans,6-trans-farnesal, 13-OxoODE, 9-cis-retinoic acid, 14,15-DiHETrE, and lithocholic acid. Additionally, five metabolites belonged to the group of organic acids and their derivatives, including acetylcysteine, ergothioneine, L-malic acid, L-phenylalanine, and 1-methylhistidine. Moreover, 18 differential metabolites were present in lower amounts in the SM group, including organic heterocyclic compounds, alkaloids and their derivatives, organic oxygen compounds, nucleosides and nucleotide analogs, benzene compounds, and linolenic acid and its derivatives.

#### 3.3.4. Kyoto Encyclopedia of Genes and Genomes Enrichment Analysis

The sample analysis of the number of differential metabolites in the SM and NO groups revealed a total of 1640 metabolites, of which 46 differential metabolites were identified, corresponding to 34 metabolic pathways (Appendix A). The bubble chart results showed that the most affected pathways (*p* < 0.05) were the citrate cycle (tricarboxylic acid (TCA) cycle) and linoleic acid, phenylalanine, retinol, pyruvate, arachidonic acid, and histidine metabolism (Figure 4A). Among the top 20 pathways with significantly differential metabolite enrichment, 6 belonged to amino acid metabolism, 5 to carbohydrate metabolism, 2 to the endocrine system, 1 to energy metabolism, 1 to immune system metabolism, 3 to lipid metabolism, and 2 to cofactor and vitamin metabolism (Figure 4B). Furthermore, a detailed examination of metabolic pathways and related differential metabolite connections revealed that arachidonic acid metabolism belonged to the lipid metabolism pathway. Within this pathway, the differential metabolite 14,15-DiHETrE demonstrated significant variation (Figure 4C). 

## 4. Discussion

Currently, there is a global annual increase in the incidence of muscle abnormalities in poultry, with different types of myopathies emerging. Specifically, there is a growing prevalence of SM in chickens [25]. It was reported that the prevalence of SM pectoralis was significantly higher in females than in males (25.0% and 3.1%, respectively). However, the reason for the higher incidence of SM in females remains unknown [26]. In this study, muscle surface examination revealed that SM-affected meat exhibited a noodle-like shape, possessed a soft texture discernable by touch, and was prone to breakage. These findings are consistent with those of Tasoniero et al. [27]. This phenomenon may be attributed to the intensive efforts made by the breeding industry to increase the growth rate of chickens, resulting in rapid growth of the pectoralis major muscles, accompanied by reduced muscle performance. The results of the meat composition test revealed that the yellowness (b*) value of the SM group was significantly higher than that of the NO group; however, the pH value and protein content were significantly lower in the SM group than in the NO group. The higher yellowness (b*) value in the SM group may be related to the accumulation of fat, given that fat exhibits a yellow hue. Moreover, SM is characterized by a large amount of fat accumulation, contributing to the yellowish appearance of the surface of the SM-affected breast muscles. This increase in yellowness aligns with the histological observations made by Ali et al., who reported a large amount of fat accumulation in the muscle fibers of SM-affected breast muscles [28], consequently explaining the elevated yellowness (b*) value. The decrease in pH observed in the SM group may be associated with changes in muscle glycogen content and alterations in glucose metabolism. However, previous studies have demonstrated an association between higher pH values and SM, WS, and WB [29,30]. The severity of myopathy can affect meat pH levels. Berri et al. suggested that the higher pH levels in myopathy-affected breast tissues stem from reduced glycolytic potential attributed to diminished carbohydrate metabolism [31]. The reduction in protein content in the SM group may be due to the degradation of myofibrillar proteins caused by the damage to the muscle fiber structure, resulting in diminished functionality of muscle proteins. This decrease in protein levels aligns with the findings of Baldi et al. [32]. This observation may be attributable to the lower hydroxylysylpyridinoline content in the intramuscular connective tissue of the surface layer of SM-affected breast muscles. 

The various myopathies discussed thus far also exhibit shared microscopic characteristics. Upon microscopic examination, the findings from this study showed that SM-affected meat was characterized by thinner, looser, and immature collagen fiber bundles that were randomly organized, forming more of a matrix. This aligns with the histopathological characteristics of SM described by Che et al. The report indicated that SM shares similar morphological changes with WB and WS, including muscle fiber degeneration, poor fiber toughness, and uniformity [33]. This phenomenon may occur because of the fragility of the connective tissue, making muscle fibers susceptible to mechanical damage during sectioning. Moreover, collagen loss or reduced performance may affect the structural integrity of the muscle. In addition, SM sections exhibited distinct features: the widespread distribution of neutrophils on the perimysium, along with severe inflammatory cell infiltration and fibrosis in the perimysium, and endomysium and abnormal adipose tissue deposition. These findings align with the results reported by Soglia et al. Loss of connective tissue, along with the deposition and infiltration of adipose tissue around the muscle, has been reported in SM tissue samples, which resulted in highly variable muscle fibers [34]. The primary driver of inflammation and severe fiber damage in the endomysium is the increase in proinflammatory cytokine levels. The substantial presence of neutrophils diffusing into the SM endomysium led to the release of cytokines, intensifying inflammation and causing muscle fiber damage. Previous studies on the inflammation formation process have reported that macrophages and satellite cells release cytokines, which prevent the proliferation of myoblasts and stimulate the accumulation of inflammatory cells, thereby affecting the growth of muscle fibers [35,36]. 

Previous reports have established that 14,15-DiHETrE, categorized under eicosanoids, constitutes a subgroup of oxidized proteins derived from arachidonic acid and polyunsaturated fatty acids [37]. It is identified as a proinflammatory class II decanoic acid, involved in inflammation and oxidative stress, potentially leading to oxidative damage in muscle cells [38,39]. The results of this study revealed a high abundance of 14,15-DiHETrE in the SM group, indicating a close association between the inflammatory phenomenon in SM and the differential metabolite 14,15-DiHETrE. Additionally, metabolomics analysis revealed that the contents of differential metabolites isotretinoin and L-malic acid were significant in the SM group. In a study by Melnik, isotretinoin was identified as an orally active retinoic acid derivative capable of inducing mutations in key regulators of apoptosis signaling, thereby increasing susceptibility to apoptosis-related adverse reactions [40]. These findings suggest that isotretinoin has the potential to stimulate muscle cell mutations, potentially contributing to muscle fiber atrophy in SM through isotretinoin-induced cell apoptosis and necrosis. L-malic acid is a water-soluble biopolymer, and the primary metabolic pathway for its biosynthesis involves the TCA cycle [41]. The TCA cycle is responsible for the complete oxidation of acetyl-CoA derived from pyruvate under aerobic conditions [42]. In this study, SM-affected broilers exhibited higher levels of the differential metabolite L-malic acid. This elevation may be attributed to reflux of the TCA cycle under hypoxic conditions, resulting in the conversion of a large amount of oxaloacetate into malic acid. Under hypoxic conditions, TCA cycle activity decreases, leading to an increase in acetyl CoA levels and a decrease in NADH and FADH_2_ quantities. The decrease in the number of these electronic carriers will weaken the inhibitory effect of fatty acid synthase, thereby promoting the synthesis of fatty acids. This hypoxic state is closely related to oxidative stress, and the combination of these two factors can contribute to the development of myopathy. This study also revealed lower levels of the differential metabolite 4-hydroxybenzaldehyde in the SM groups. Chen et al. demonstrated that 4-hydroxybenzaldehyde has strong antioxidant activity and inhibits cell migration and proliferation by mitigating oxidative stress and activating nitric oxide synthase expression, thereby reducing inflammation and muscle degeneration [43]. The reduced level of 4-hydroxybenzaldehyde in SM may result in a downregulated ability to counteract oxidative stress and an inability to exert anti-inflammatory effects, allowing inflammatory cells to persistently infiltrate muscle fibers. 

This study presents findings from research into related metabolic pathways. Differential metabolites play a role in multiple metabolic pathways, among which linoleic acid and arachidonic acid metabolism are anticipated to be most affected by SM myopathy. Linoleic acid, the most commonly consumed polyunsaturated fatty acid in human diets, can undergo esterification to form neutral and polar lipids [44]. Linoleic acid can be desaturated to form arachidonic acid, which is further converted into various bioactive compounds known as eicosanoids [45]. The results of this study showed that the most significant differential metabolite in the arachidonic acid metabolism pathway was 14,15-DiHETrE. This result may be attributed to the fact that 14,15-DiHETrE is an eicosanoic acid. Elevated levels of 14,15-DiHETrE can lead to the activation of numerous proinflammatory factors, neutrophils, and macrophages, thereby causing lesions in the pectoralis major muscle. In addition, metabolic pathways related to amino acid, phenylalanine, and histidine metabolism were abundant in the NO and SM samples. Bazzichi et al. demonstrated an association between SM-affected broiler chickens and amino acid metabolism disorders [46]. This observation indicates a close connection between inflammation in SM and an increase in phenylalanine levels. Higher levels of phenylalanine metabolism correspond to an increased likelihood of inflammation and infection. Histidine, an essential amino acid, can limit carnosine synthesis, which exerts a strong antioxidant effect [47]. Branco et al. reported that histidine undergoes decarboxylation to form histamine under the action of histidine decarboxylase, and the activation of the histamine receptor subsequently leads to an increase in the number of eosinophils and neutrophils at the inflammatory site [48]. The abundance of histidine metabolism in SM indicates that its prevalence in this pathway may exacerbate the development of SM myopathy, trigger cellular inflammation, and cause muscle cell relaxation. Due to the limited research on the metabolic characteristics of SM and the limited reference literature available, it may lead to a lack of comprehensive research on the factors affecting the pathogenesis of SM deficiency. Future research should elucidate the metabolic status of affected muscles and the effects of myopathy in defect situations.

## 5. Conclusions

In summary, in this study, we systematically compared the differences in appearance, meat quality characteristics, muscle fiber characteristics, and metabolic components between SM and NO tissues. Compared to the NO group, the SM group exhibited a softer and more brittle meat texture with a strip-shaped surface. The yellowness and brightness observed in the SM group were significantly higher than those in the NO group; however, the pH and protein levels were significantly lower than those in the NO group. Microscopic observation revealed an abundance of neutrophils in SM perimysium, accompanied by inflammatory necrosis in the muscle cells. The differential metabolites 14,15-DiHETrE, isotretinoin, and L-malic acid were abundant in the SM and NO groups. Moreover, some of these metabolites are implicated in inflammation and oxidative stress, capable of inducing apoptosis and promoting muscle cell necrosis. We infer that the key differential metabolites 14,15-DiHETrE and L-malic acid and crucial metabolic pathways, including linoleic acid, arachidonic acid, phenylalanine, and histidine metabolism, which are related to inflammation and oxidative stress, are implicated in SM. These findings offer new insights into the muscle characteristics of SM and contribute to the identification of potential mechanisms and nutritional regulatory targets for SM myopathy. Due to the importance of myopathy in the composition and function of meat, the impact of myopathy on muscle integrity and intrinsic characteristics (i.e., degeneration, inflammation, and oxidative stress) deserves further research.

## Figures and Tables

**Figure 1 animals-14-01236-f001:**
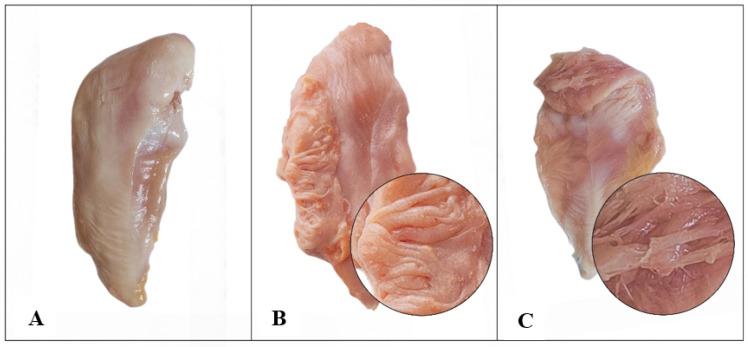
Muscle quality level. (**A**) Normal (NO) muscle. (**B**,**C**) Spaghetti meat (SM).

**Figure 2 animals-14-01236-f002:**
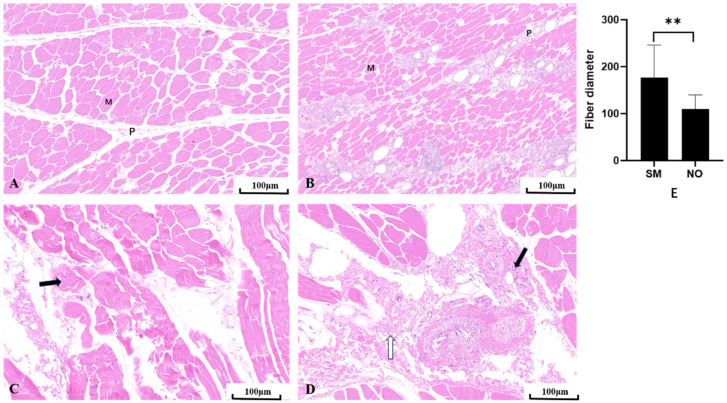
Breast muscle tissues. (**A**) Normal (NO) breast muscle. (**B**) Spaghetti meat (SM) muscle; M = muscle fiber, P = perimysium. Scale rod, 100 μm. (**C**) Myofiber fragmentation (black arrow). (**D**) Inflammatory cells (black arrow) and fibrotic (white arrow) interstitial changes. (**E**) Comparison between SM and NO fiber diameters. Six parts of each section were located and observed under a 20× field of view; ** indicates an extremely significant difference (*p* < 0.01).

**Figure 3 animals-14-01236-f003:**
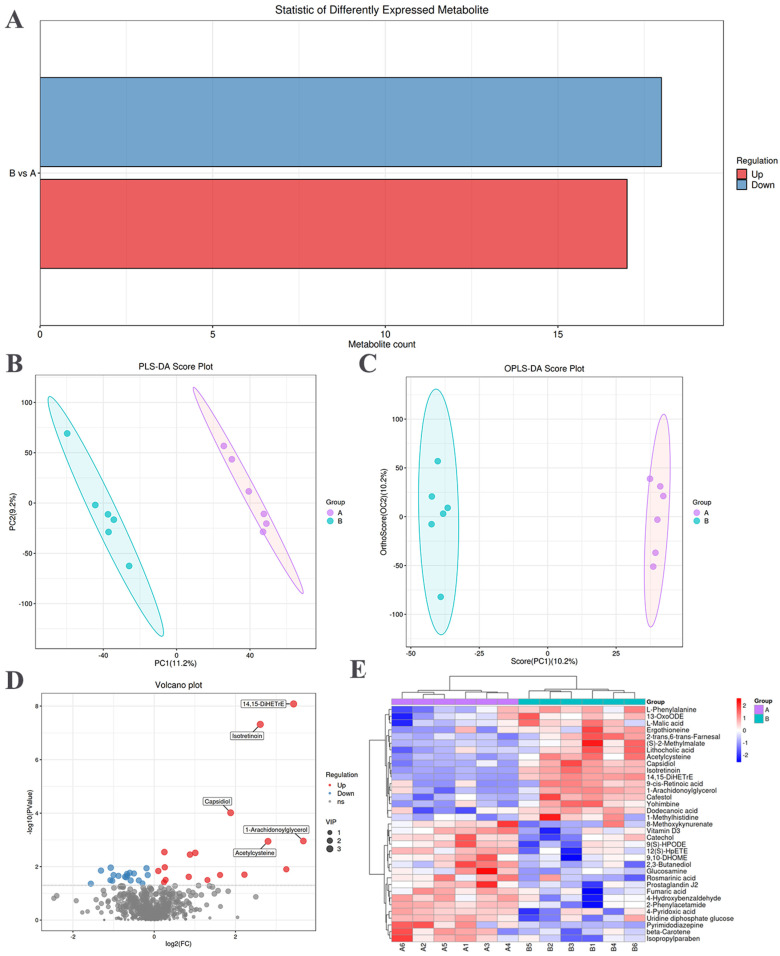
Differential metabolites related to spaghetti meat (SM) and normal (NO) tissues. (**A**) Statistical histogram of differential metabolites between the NO and SM groups. (**B**) Metabolite latent structure partial least squares discriminant analysis (PLS−DA) score plot. (**C**) Orthogonal corrected PLS−DA (OPLS−DA) score plot of metabolites. (**D**) Volcano plot of tissue metabolites. Each dot represents a metabolite, with red and blue dots indicating up and downregulated metabolites, respectively. Gray dots indicate metabolites with no differences. (**E**) Hierarchical clustering analysis and heat map display of 35 identified metabolites.

**Figure 4 animals-14-01236-f004:**
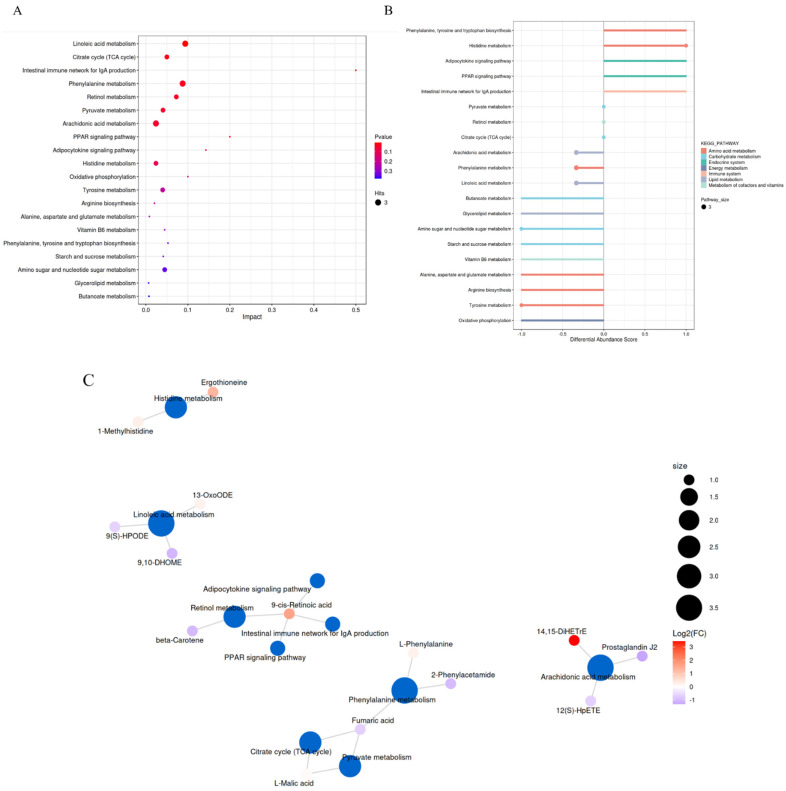
Metabolic pathways of differential metabolites between spaghetti meat (SM) and normal (NO) tissues. (**A**) Bubble chart of the influencing factors of metabolic pathways. Each point represents a metabolic pathway; the abscissa indicates the impact value enriched into different metabolic pathways, and the ordinate indicates the enriched pathway. The dots indicate the corresponding number of metabolic molecules on the pathway. (**B**) Differential enrichment score chart. The abscissa represents the discriminant analysis (DA)−score value, calculated as follows: DA−score = (number of upregulated substances−number of downregulated substances)/total number of differential substances in the pathway. The ordinate indicates the metabolic pathway, and the size of the column vertex represents the number of differential metabolites enriched in the pathway. (**C**) Metabolic pathways and related differential metabolite network diagram. Blue points represent pathways, and other points represent metabolites. The size of a pathway point indicates the number of metabolites, with a greater number of connected metabolites corresponding to a larger point.

**Table 1 animals-14-01236-t001:** Detection of quality indicators in spaghetti meat (SM)-affected and unaffected normal (NO) breast muscle tissues.

Traits	Muscle Condition (M)		*p*-Value
	NO	SM	
L*	45.46 ± 3.86 ^b^	47.85 ± 3.58 ^a^	*p* = 0.02
a*	1.61 ± 0.70	1.87 ± 1.27	*p* = 0.33
b*	7.93 ± 1.52 ^b^	9.08 ± 2.83 ^a^	*p* = 0.04
pH	5.84 ± 0.33 ^a^	5.63 ± 0.17 ^b^	*p* = 0.04
Hydraulic system, n	0.24 ± 0.05	0.25 ± 0.06	*p* = 0.87
Shear force, %	27.48 ± 6.25 ^a^	18.42 ± 6.78 ^b^	*p* = 0.00
Moisture, %	72.75 ± 0.56	72.48 ± 0.72	*p* = 0.16
Protein, %	24.93 ± 0.54 ^a^	24.19 ± 0.44 ^b^	*p* = 0.00
Fat, %	1.45 ± 0.43	1.82 ± 0.79	*p* = 0.06
Collagen ^1^	0.54 ± 0.10	0.59 ± 0.11	*p* = 0.08
Carcass weight, g	1864 ± 213.20	1857 ± 209.70	*p* = 0.98
Unilateral breast, g	138.41 ± 13.42	136.78 ± 12.95	*p* = 0.96

^a, b^ Means within the same row followed by different superscripts differ *p* < 0.05. ^1^ mg/g of muscle tissue.

**Table 2 animals-14-01236-t002:** Identification of significantly different metabolites in the tissues of spaghetti meat (SM)-affected and unaffected normal (NO) broiler chickens.

Metabolites	MZ	RT (min)	VIP	*p*	FC
Upregulated					
(S)-2-Methylmalate	149.06	258.30	2.23	0.02	4.65
Acetylcysteine	163.00	408.50	2.74	0.00	6.99
Dodecanoic acid	199.98	989.60	2.20	0.02	1.80
Capsidiol	219.17	648.00	2.75	0.00	3.68
2-trans,6-trans-Farnesal	221.19	918.40	2.35	0.00	1.84
Ergothioneine	230.10	108.10	1.92	0.03	2.48
13-OxoODE	277.22	810.50	1.88	0.03	1.18
Isotretinoin	301.22	818.70	3.04	0.00	6.10
9-cis-Retinoic acid	301.22	839.30	2.03	0.02	3.08
Cafestol	317.21	870.20	2.42	0.00	1.19
14,15-DiHETrE	321.24	743.40	3.07	0.00	10.75
Lithocholic acid	377.31	797.60	2.12	0.01	9.49
1-Arachidonoylglycerol	379.28	935.00	2.56	0.00	12.73
L-Malic acid	133.02	123.90	2.19	0.01	1.07
L-Phenylalanine	164.07	300.60	2.21	0.01	1.20
1-Methylhistidine	168.08	107.30	2.01	0.03	1.21
Yohimbine	353.18	740.80	2.48	0.00	2.02
methylnaphthaleneacetic acid					
Catechol	111.02	909.00	1.98	0.03	0.77
4-Hydroxybenzaldehyde	123.04	204.50	2.09	0.02	0.66
2-Phenylacetamide	136.08	204.30	2.20	0.02	0.50
2,3-Butanediol	154.99	59.90	2.06	0.02	0.61
Glucosamine	180.09	358.40	2.08	0.02	0.54
8-Methoxykynurenate	220.06	490.50	1.90	0.04	0.82
Pyrimidodiazepine	221.09	424.80	2.39	0.01	0.48
9(S)-HPODE	313.24	701.10	2.09	0.02	0.63
12(S)-HpETE	319.23	867.70	2.05	0.02	0.63
Rosmarinic acid	360.24	462.70	2.07	0.02	0.71
Vitamin D3	384.35	913.40	2.11	0.02	0.89
beta-Carotene	536.16	919.00	1.90	0.03	0.49
Fumaric acid	115.00	74.90	2.09	0.02	0.62
Isopropylparaben	179.07	741.80	2.23	0.01	0.87
4-Pyridoxic acid	182.05	340.20	2.09	0.04	0.34
9,10-DHOME	314.24	827.40	1.93	0.03	0.47
Prostaglandin J2	315.19	682.90	2.19	0.01	0.40
Uridine diphosphate glucose	565.05	80.50	2.17	0.03	0.67

Note: FC = fold change, the average peak area obtained by the SM group/the average peak area obtained by the NO group. An FC value exceeding 1 indicates the presence of more metabolites in the SM group compared to the NO group. *p* values were obtained via *t*-test.

## Data Availability

All data generated or analyzed during this study are included in this published paper.

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
