# Peer review of "Broiler Spaghetti Meat Abnormalities: Muscle Characteristics and Metabolomic Profiles"

_animals, 2024, doi:10.3390/ani14081236_

Round 1
Reviewer 1 Report
Comments and Suggestions for Authors
This manuscript has certain significance, but there are still some issues, as follows:
1. Abstract: this part describes too much, please refine it.
2. Line 54-56, 224-226, 347-349: Please add references.
3. Materials and Methods: The test design is not clear, as the grouping situation, how to sample.
4. Line 107: 49? Please explain the basis for selection. The sex of the chosen chicken? Why? Where do they come from? Please explain these clearly.
5. Line 133: Please note whether Spaces are required between numbers and units. This problem also exists elsewhere in the manuscript. Please check the manuscript carefully.
6. Line 185: “......interspersed with numerous rice-like yellow substances.” rice-like yellow substances? This description does not fit, just from the picture cannot tell.
7. Line 194-196, 293-295: P needs italics.
8. Figure 3E: The difference between the groups is very obvious, is the data processing unreasonable?
9. Line 265-268: This description is recommended for inclusion in the text.
10. Line 337: There is an error in the reference.
11. Line 366-367: Is there a problem with this part of the description? Suggest the author rethink. TCA cycle is affected, but the oxidation of fatty acids is not affected, fatty acids will still oxidize to acetyl CoA, resulting in excess acetyl CoA, resulting in the formation of a large number of ketone bodies.
12. Table 1, some indicators have not been statistically analyzed, please add.
13. Table 1, 2: please unify the digits after the decimal point.
14. Figure 3 and Figure 4: the definitions of the figures are too low to see clearly, please provide them again.
15. References: more than 50% of the references are not the last five years of reference, it is recommended that the author replace some old references with newer references.
16. The grammatical tenses of the whole manuscript are problematic, and the quality of English writing needs to be improved.
Author Response
请参阅附件。

Reviewer 2 Report
Comments and Suggestions for Authors
1. Terms must be used consistently: spaghetti meat (SM) is an abnormality shown in the muscles commonly called "breast muscles". In this manuscript this term is used 26 time, but 10 times the term "chest muscles" is used, and 7 times the term "pectoralis muscles" is used. This inconsistency is inappropriate, and must be corrected.
2. There are other unclear terms, e.g. "current slaughterhouse" (line 100). What is it?
3. In Materials and Methods, section 2.3, section 2.2, it must be stated if the 200 broilers were males or females or mixed. This is critically important in view of the fact mentioned in lone 71.
4. In Materials and Methods, section 2.3, the slaughtering procedures must be described in full details, because some procedures (e.g. defeathering, chilling) can affect the incidence and severity of SM.
5. Also in this section, in addition to age of slaughter, mean body weight at slaughter must be presented. Moreover, the mean body weight of the 18 NO broilers and the 18 SM broilers should be presented, in case they differ in body weight (and/or in gender).
6. In this section (line 104) it is written that two levels of SM were scored, mild (SM1) and severe (SM2), and these two levels are shown in Figure 1. However, already from line 105, and throughout the entire manuscript, only one group of 18 SM samples is mentioned… It must be stated clearly if these 18 samples were SM1 or SM2 or a mixture of both. Actually, the study should measure 3 groups of samples (NO, SM1, SM2) rather than two groups only.
7. What were the proportions of NO, SM1 and SM2 among the 200 tested broilers?
8. The following two papers must be included in the literature review and in the discussion of the results.
Bailey RA (2023), Strategies and opportunities to control breast myopathies: An opinion paper.
Front. Physiol. 14:1173564. doi: 10.3389/fphys.2023.1173564
Bailey, R. A., Souza, E., and Avendano, S. (2020). Characterising the influence of genetics on breast muscle myopathies in broiler chickens. Front. Physiol. 11, 1041. doi:10.3389/fphys.2020.01041
Comments on the Quality of English Language1. Terms must be used consistently: spaghetti meat (SM) is an abnormality shown in the muscles commonly called "breast muscles". In this manuscript this term is used 26 time, but 10 times the term "chest muscles" is used, and 7 times the term "pectoralis muscles" is used. This inconsistency is inappropriate, and must be corrected.
2. There are other unclear terms, e.g. "current slaughterhouse" (line 100). What is it?
Round 2
Reviewer 1 Report
Comments and Suggestions for Authors
The author has made revisions to the manuscript according to the reviewer's comments, which meet the requirements for publication. I suggest accepting and publishing it.
Author Response
Response to Reviewer 1 Comments
Comments and Suggestions for Authors
The author has made revisions to the manuscript according to the reviewer's comments, which meet the requirements for publication. I suggest accepting and publishing it.
Response: We are grateful to the reviewers for their insightful and constructive comments. These suggestions have further enhanced the quality of this manuscript.

Reviewer 2 Report
Comments and Suggestions for Authors
1. Females are more prone to SM than males. This discrepancy in the experimental design cannot be corrected, but at least it should be mentioned and discussed.
2. It is not clear if there were only 18 broilers with SM, and if all the other 182 broilers were free of any level of SM. Moreover, it is not clear if the classification to either SM or NO was done on whole birds, or on deboned unilateral breast muscles. This is very important!!!
3. The classification of SM versus NO breast muscles must be detailed! It is well known that the level of SM gradually changes from NO to severe SM. The manuscript must refer to this aspect.
4. What was the body weight of the SM broilers versus the NO broilers? Without this information, it is possible that the body weight of SM broilers was significantly different from that of the NO broilers, and consequently observed differences were related to body weight and not to SM.
Either the whole carcasses or unilateral breast muscles were classified as SM or NO, these weights must ber analyzed and presented in Table 1.
The language must be checked and corrected.
